# Role of Phosphorylation of Serotonin and Norepinephrine Transporters in Animal Behavior: Relevance to Neuropsychiatric Disorders

**DOI:** 10.3390/ijms26167713

**Published:** 2025-08-09

**Authors:** Lankupalle D. Jayanthi, Sammanda Ramamoorthy

**Affiliations:** Department of Pharmacology and Toxicology, Virginia Commonwealth University, Richmond, VA 23298, USA

**Keywords:** biogenic amine transporters, phosphorylation, regulation, transgenic animals, neuropsychiatric behaviors

## Abstract

Serotonin and norepinephrine transporters (SERT and NET), located on the presynaptic terminals, regulate serotonergic (5-HT) and noradrenergic (NE) neurotransmission by rapid reuptake of released amines from the synapse. Clinically used antidepressants and highly abused psychostimulants have high affinity for these transporters. The function and expression of SERT and NET are altered in mood disorders and psychostimulant use. Therefore, appropriate functional regulation of SERT and NET is important in maintaining normal homeostasis of 5-HT and NE signaling. Both SERT and NET possess kinase-specific phospho-sites/motifs and exist in phosphorylated state. Several cellular protein kinases and phosphatases regulate the dynamics of phosphorylation of SERT and NET, which in turn determine the subcellular expression and trafficking, microdomain-specific protein–protein interactionsprotein-protein interactions, transporter protein degradation and ultimately transport capacity. Dysregulations in the dynamics of SERT and NET phosphorylation and their impact on functional regulation might contribute to neuropsychiatric disorders. However, the neurobiological consequences and behavioral outcome of SERT and NET phosphorylation in vivo are not fully understood. Studies using intact animal models that directly link the phosphorylation of SERT and NET to regulatory molecular mechanisms and animal behavior are just beginning to emerge. This review summarizes our understanding of the role of phosphorylation-dependent regulation of SERT and NET in animal behaviors relevant to mood and psychostimulant use disorders. Understanding of phosphorylation-dependent molecular mechanisms of SERT and NET regulation is pivotal to identifying potential candidate mechanisms as therapeutic targets in the treatment of neuropsychiatric disorders.

## 1. Introduction

Overview of Serotonin and Norepinephrine Transporters and Their Regulation: Biogenic monoamine (MA) neurotransmitters, including serotonin (5-hydroxytryptamine, 5-HT) and noradrenaline/norepinephrine (NA/NE), are synthesized in vivo or de novo from tryptophan and tyrosine, respectively. Each amine regulates distinct behavioral and physiological functions in both the central and peripheral nervous systems. 5-HT is involved in modulating mood, aggression, motivation, appetite, sleep, cognition and sexual activity and altered 5-HT signaling is involved in mental illnesses [1,2,3,4]. 5-HT also regulates vasoconstriction, gastrointestinal and placental function. NE controls arousal, mood, attention, stress-response and affective disorders [5,6,7,8]. NE is also the major neurotransmitter in the sympathetic nervous system and controls cardiovascular functions [9,10,11,12,13,14]. Many macromolecules such as biosynthetic enzymes, secretory proteins, ion channels, pre- and postsynaptic receptors and transporters regulate serotonergic and noradrenergic signals.

The presynaptic plasma membrane transporters for serotonin (SERT) and norepinephrine (NET) regulate extracellular neurotransmitter levels by rapidly clearing released neurotransmitters from the extracellular space. SERT (SLC6A4) and NET (SLC6A2) belong to the same gene (SLC6) family. Both SERT and NET display a predicted structure with 12 transmembrane domains, featuring intracellular amino and carboxy termini. This topology has been confirmed by the high-resolution crystal structures of human (hSERT) [15] and hNET [16] as well as bacterial homologue, LeuT, of the mammalian monoamine transporters [17].

Clinically used antidepressants and widely misused psychostimulants target MA transporters [18,19]. Deletion of gene encoding SERT or NET in mice resulted in interdependent mechanisms of amine biosynthesis, amine storage, receptor sensitivity and transporter expression [20,21,22,23,24,25,26,27]. The implications of SERT and NET in regulating serotonergic and noradrenergic neurotransmission, their roles in disease processes, and their targets for various therapeutically active drugs and substances of abuse highlight the importance of understanding the underlying cellular and molecular mechanisms. These mechanisms involve how various signaling pathways modify SERT and NET, altering their functional properties and trafficking. Recent advances in the field have provided a wealth of knowledge on how the expression and functional properties of SERT and NET are regulated at both the gene and protein levels.

SERT and NET functional expression can be regulated through both phosphorylation-dependent and phosphorylation-independent post-translational modifications. Phosphorylation at specific sites on SERT and NET leads to changes in their intrinsic transporter properties, stability, surface expression, and the residency time of transporter proteins by influencing the exocytic fusion of transporter-containing vesicles with the plasma membrane and their sequestration from the membrane through regulation of endocytic pathways. Additionally, transporter regulation can also occur through their association with other interacting proteins and redistribution in membrane microdomains, either via phosphorylation-dependent or -independent pathways [28].

This review discusses an important molecular post-translational mechanism named the phosphorylation of SERT and NET that affects not only several key properties of these transporters but also their role in animal behavior. Some earlier aspects of monoamine transporter phosphorylation and regulation, including the phosphorylation of dopamine (DA) transporter (DAT), are detailed in several excellent reviews [28,29,30,31,32,33,34,35,36]. We review here the studies on SERT and NET that have explored the relationship between changes in transporter phosphorylation states and transporter function. We also review recent progress made in the functional analysis of phospho-defective SERT and NET expressed in genetically modified mice, focusing on their roles in human disease and animal behavior. The review emphasizes the role of endogenous phospho-SERT and phospho-NET in animal behavior with the intent of shining a light on the underlying compromised neuronal mechanisms to help us better understand the etiology of mental illnesses and psychostimulant substance use disorder (SUD), and to open the door for developing alternative pharmacological strategies targeting specific regulatory motifs for treating neuropsychiatric disorders such as depression and SUD.

## 2. The Serotonin Transporter Phosphorylation and Its Significance

### 2.1. Regulation of SERT by Phosphorylation

The uptake of synaptic 5-HT through Na^+^/Cl^−^-dependent SERT is the primary process of terminating serotonergic neurotransmission. Human SERT encodes a protein with 630 amino acids, featuring 12 hydrophobic transmembrane domains (TMDs), two canonical sites for N-linked glycosylation between TMDs 3 and 4, and intracellular amino and carboxyl terminals [37,38]. Many canonical phosphorylation sites, including serine (Ser), threonine (Thr), and tyrosine (Tyr) for various protein kinases are present in the cytoplasmic domains of SERT [37,39,40]. Research has documented kinase-mediated phosphorylation of SERT and its regulation by substrates and ligands [41,42,43]. Furthermore, several cellular protein kinases, such as protein kinase B (Akt), calcium/calmodulin-dependent kinase II (CaMKII), glycogen synthase kinase-3 beta (GSK3β), protein kinase C (PKC), p38 MAPK, cGMP-dependent kinase (PKG), and Src kinase, dynamically regulate SERT function, trafficking, stability, and phosphorylation [36,44,45,46,47,48,49,50,51,52,53,54,55,56,57,58,59,60]. Additionally, auto- and hetero-receptors expressed on serotonergic neurons regulate intracellular kinase and phosphatase signaling pathways that modulate SERT function and 5-HT signaling [36,61]. Therefore, second messenger-kinase and phosphatase signaling pathways activated by presynaptic receptors are essential for regulating SERT function through phosphorylation and dephosphorylation processes.

PKC-mediated SERT phosphorylation and downregulation are blocked by 5-HT and other SERT substrates and this blockade can be prevented by SERT inhibitors or by removal of Na^+^ and Cl^-^ ions which are required for 5-HT transport [43]. These findings suggest that expression of SERT on the plasma membrane might be tightly linked to the activity state of SERT in that substrate-induced conformational changes in SERT obscure the PKC sites within the SERT protein [43]. Peripheral tissues such as platelets express SERT and decreased platelet SERT binding sites are evident in depression [62,63]. Jayanthi et al., reported that PKC activation regulates endogenous SERTs expressed in platelets in a biphasic manner [47]. In the initial phase, PKC activation inhibits SERT activity by decreasing its affinity for 5-HT and transport velocity without altering surface SERT levels. In the later phase, transport velocity and surface SERT levels are reduced due to increased endocytosis. This biphasic inhibition of SERT is linked to the sequential phosphorylation of SERT on the plasma membrane. Initially, phosphorylation occurs at Ser residues, causing changes to the intrinsic properties or silencing of the SERT. Later, phosphorylation at Thr residues prompts the internalization of the phosphorylated SERT. While studies have shown that mutating Thr-276 and Ser-277 to non-phosphorylatable Ala prevents the early PKC-induced reduction in SERT function [64], there is currently no direct evidence to confirm the phosphorylation of these residues or any others that are necessary for PKC-mediated regulation of SERT endogenously. Unlike PKC-dependent downregulation of SERT, p38 MAPK-mediated SERT upregulation involves enhanced SERT surface expression and SERT phosphorylation [46] suggesting that PKC and p38 MAPK phosphorylate SERT at distinct sites to regulate its function and trafficking differentially. Identifying the specific sites of phosphorylation mediated by PKC and p38 MAPK will shed light on the cellular and molecular aspects of SERT regulation in response to the activation of these kinases. Since p38 MAPK is a stress-activated kinase, its regulation of SERT may serve as a presynaptic mechanism to help maintain proper synaptic 5-HT levels, regulating serotonergic transmission during stressful situations and aiding in stress coping.

Published findings reveal that activation of PKG stimulates SERT activity via trafficking-dependent and independent pathways [40,45,57]. Ramamoorthy et al., found that activation of PKG phosphorylates endogenous SERT in the rat midbrain and hSERT expressed in CHO-1 cells [45]. In this study, they identified a phosphorylation site in SERT and demonstrated that phosphorylation of Thr-276 is essential for PKG-mediated regulation of SERT. Using phosphoamino acid analysis of phosphorylated native SERTs from rat midbrain, they reported that only Thr residues on SERT are phosphorylated following PKG activation by cGMP. By replacing Thr-276 with the non-phosphorylatable alanine in hSERT, they showed elimination of cGMP-mediated stimulation of 5-HT transport and SERT phosphorylation. Phosphorylation of Thr-276 adds a negative charge that could influence SERT activity. Supporting this notion, replacing Thr-276 with aspartic acid (Asp), which in its ionized form carries a negative charge, would mimic PKG effects on 5-HT uptake. In fact, by mutating Thr-276 to Asp, which simulates phosphorylation, the investigators found increased 5-HT uptake to a level similar to cGMP-stimulated 5-HT uptake in wild-type hSERT-expressing cells, and the uptake was no longer affected by cGMP stimulation. These findings mark the first identification of a phosphorylation site in hSERT and demonstrate that PKG phosphorylates hSERT on Thr-276, resulting in increased 5-HT uptake. Zhang et al. demonstrated that inward-open stabilizing agents, such as the SERT substrate 5-HT and inhibitor ibogaine, increase the phosphorylation of Thr-276. In contrast, agents that stabilize outward-open conformations, such as Na+ and cocaine, reduce SERT Thr-276 phosphorylation [51]. This dynamic rearrangement of SERT conformation regulates the phosphorylation at the Thr-276 site, thereby allowing PKG to enhance the catalytic activity of SERT in transporting 5-HT.

Besides the role of Ser/Thr protein kinases in regulating SERT, evidence indicates that a family of tyrosine protein kinases also controls SERT activity and phosphorylation. Studies by Zarpellon et al., and others have demonstrated that while Src kinase inhibitors decrease 5-HT uptake and tyrosine phosphorylation of SERT, tyrosine protein phosphatase inhibitors increase SERT tyrosine phosphorylation and SERT activity in human platelets [55,65]. Furthermore, Zarpellon et al., provided evidence for SERT and Src existence as a complex, as shown by the coimmunoprecipitation approach. Using a heterologous expression model of human placental trophoblasts, Annamalai et al., demonstrated that expressing the tyrosine kinase Src increased SERT function, tyrosine phosphorylation, and SERT protein expression while knocking down Src via siRNA inhibited SERT activity [66]. The amino acid sequence of SERT contains four intracellular tyrosine residues (Tyr-47, Tyr-142, Tyr-350, and Tyr-358) that are potential phosphorylation sites for tyrosine kinases [37]. Site-directed mutation of Tyr-47 and Tyr-142 to non-phosphorylatable phenylalanine diminished the Src-induced increase in 5-HT transport, SERT protein expression, and SERT-tyrosine phosphorylation. However, single-site mutation of Tyr-350 or Tyr-358, or mutation of all four tyrosine residues (Tyr-47, Tyr-142, Tyr-350, and Tyr-358) to phenylalanine, completely abolished SERT expression. Furthermore, this study also demonstrated that inhibiting protein tyrosine kinase promotes degradation of SERT protein and reduces overall SERT protein levels. These findings establish a direct causal relationship between the phosphorylation of Tyr-47, Tyr-142, Tyr-350, and Tyr-358 in SERT and the regulation of SERT stability and function.

Heteroreceptors expressed in 5-HT terminals regulate SERT function and phosphorylation. Activation of histamine receptor (H_3_R) decreases SERT function, which is accompanied by reduced surface expression and phosphorylation via the CaMKII/calcineurin pathways [52]. Kappa opioid receptor (KOR) agonism triggers SERT phosphorylation, inhibits SERT function, and decreases surface SERT expression through CaMKII and Akt downstream signaling [67]. These findings indicate that the neurobiological and behavioral effects of histamine and dynorphin (the endogenous ligand for KOR) may result from coordinated regulation of 5-HT release and SERT-mediated 5-HT clearance, influencing serotonergic neurotransmission. Figure 1 summarizes kinase-, phosphatase- and receptor-mediated regulation of SERT activity by phosphorylation at specific phospho-sites.

### 2.2. Role of SERT Phosphorylation in Mood Disorders

Altered SERT binding and SERT gene linkages have been linked to depression and anxiety disorders [2,68,69]. Notably, a meta-analysis of molecular imaging of SERT and biochemical post-mortem findings from depression patients revealed reduced SERT in the amygdala and enhanced binding in the hippocampus [70,71]. Altered hippocampal-amygdala function is consistently implicated in human depression and anxiety, as well as impaired learning, stress responses, and social behaviors, with direct connections between these brain nuclei. Patients with depressive disorder often also experience cognitive dysfunctions, which manifest in increased anxiety and memory deficits [72,73]. While 5-HT depletion induces mood and memory dysfunction [74,75], selective serotonin reuptake inhibitor (SSRI)-treated depressed patients show increased recognition of memory as well as alleviated anxiety and depressive mood [76]. Moreover, mice lacking SERT exhibit mood and memory dysfunction [77,78]. These findings indicate a pathophysiological role for SERT in hippocampal-amygdala functions related to anxiety, memory/learning, and depression, and therefore, targeting SERT might help alleviate these psychiatric symptoms. Glycogen synthase kinase 3ß (GSK3ß) in the hippocampus and striatum plays a role in the regulation of behaviors evaluated in rodent models of anxiety and depression [79]. Pharmacological inhibition of GSK3 or neuron-specific deletion of GSK3ß in 5-HT neurons produces effects similar to those of antidepressants [80,81,82]. SERT inhibitors regulate the activity of GSK3ß, and variants in the GSK3ß promoter, as well as increased GSK3 activity, are associated with depression, suicide, mood disorders, and various neurological disorders [79,83]. Moreover, interactions between GSK3β promoter variants and hSERT influence clinical responses to antidepressants. As a result, abnormal regulation of GSK3-mediated SERT may occur in neuropsychiatric disorders. Ragu Varman et al. found that active GSK3β reduced SERT activity and surface expression while enhancing SERT phosphorylation [53]. A mutation of consensus GSK3 site Ser-48 to non-phosphorylatable Ala in hSERT caused a complete absence of GSK3β-mediated SERT inhibition and phosphorylation, demonstrating the causal and mechanistic relationship between SERT-Ser-48 phosphorylation and GSK3β-mediated SERT regulation. While additional translational studies are needed to connect behaviors with GSK3β-mediated SERT-Ser-48 phosphorylation, these findings suggest that GSK3β functions as a signal integrator for 5-HT clearance by modulating SERT activity, serotonergic neurotransmission, and behaviors. This indicates that dysregulated GSK3, linked to psychiatric disorders, may disrupt SERT phosphorylation and 5-HT clearance, which in turn can lead to a shift towards maladaptive behaviors resulting in psychiatric disorders.

It has been shown that rare human SERT coding variants identified in human subjects diagnosed with obsessive–compulsive disorder (OCD) and autism spectrum disorder (ASD) impact transporter phosphorylation, cell surface trafficking and/or conformational dynamics [84]. Prasad et al., have identified that naturally occurring hSERT coding variants Ile425Val and Gly56Ala associated with OCD and ASD, respectively, are shown to be not further responsive to PKG and p38 MAPK activation [85]. Later, studies by Veenstra-VanderWeele et al., and Siemann et al., have shown that knock-in mice carrying Gly56Ala mutation in SERT exhibit impaired social behavior, communication as well as abnormal repetitive behaviors which represent core deficits in ASD [86,87]. Basal or constitutive 5-HT clearance in the hippocampus is elevated in these mice, along with hyperserotonemia and hyperphosphorylation of SERT that is sensitive to p38 MAPK inhibition. It is postulated that elevated SERT constitutive phosphorylation, caused by the Ala56 mutation, might prevent this variant from switching between high and low activity states. Indeed, is important to note the study by Robson et al., where repeated administration of MW 150, a CNS-penetrant p38 MAPK inhibitor reverses these behavioral impairments along with normalization of SERT function in the hippocampus highlighting the potential therapeutic use of this kinase inhibitor in the treatment of ASD [88]. To understand how SERT phospho-sites are exposed during SERT conformational changes from low to high or high to low active state, another study by Chan et al., showed that the Thr276 residue is exposed in a conformation-dependent manner using molecular modeling [89]. This study also demonstrated the significance of SERT-Thr276 phosphorylation using knock-in mice carrying Thr276Ala mutation in that the Thr276Ala mutant mice exhibit sex-specific differences in repetitive and social interactions. Kilic et al. reported that expressing the hSERT Ile-425Val variant in cell models demonstrates higher HT uptake activity, which is unresponsive to the ability of PKG to stimulate SERT activity [90]. Furthermore, mutating the PKG phosphorylation site Thr-276 to Ala in the hSERT Ile-425Val variant eliminated the inhibitory effects of PKG observed in the Ile-425Val variant. Additionally, the heightened activity observed in the Ile-425Val variant is comparable to the level of the phosphomimetic form of hSERT (Thr-276Asp) [56], suggesting that the Ile-425 to Val mutation found in OCD may influence SERT conformations toward a sustained and elevated PKG-mediated Thr-276 phosphorylation, facilitating increased 5-HT clearance and perhaps OCD associated behaviors. While it is important to emphasize that the Gly56Ala and Ile-425Val coding variants of hSERT are rare and have been detected in families with OCD and ASD, caution is necessary when considering the implications of these findings in the genetics of these disorders and their potential causative roles.

## 3. The Norepinephrine Transporter Phosphorylation and Its Significance

### 3.1. Regulation of NET by Phosphorylation

The termination of noradrenergic neurotransmission occurs through the rapid reuptake of released synaptic NE by Na^+^/Cl^−^ dependent NET which shares high homology with SERT. The human NET consists of 617 amino acids and contains 12 hydrophobic transmembrane domains (TMDs) with a cytoplasmic amino terminus and a carboxy terminus [91]. Like SERT, human and murine NETs contain a large hydrophilic loop encompassing TMDs 3 and 4, containing canonical N-linked glycosylation sites and several potential phosphorylation sites in intracellular loops for several kinases [91,92]. The NET protein exists in phosphorylated form and NET function and phosphorylation are regulated by kinases, receptors and psychostimulants [93,94,95] and previously reviewed by us [36]. Downregulation of NET function involves internalization of NET protein from the plasma membrane [93,96]. Activation of muscarinic acetylcholine receptors by the agonist methacholine in transformed SK-N-SH cells that endogenously express NET and muscarinic acetylcholine receptors triggers NET downregulation [96,97]. The inhibition of PKC eliminates PKC-mediated effects but only partially blocks the effects mediated by methacholine. The activation of PKC and pathways that depend on the mobilization of Ca^2+^ from intracellular stores are involved in the regulation of NET by muscarinic receptors [96,97]. Together, these data suggest the presence of PKC-independent regulatory pathways that maintain NET surface expression and/or intrinsic catalytic activity of the NET. One such pathway is phosphorylation of NET protein and the first evidence showing NET as a phospho-protein comes from studies using rat placental trophoblast where PKC activation enhances NET phosphorylation [93]. Jayanthi et al. provided key evidence for lipid raft-mediated endocytosis serving as the mechanism for the PKC-mediated downregulation of NET through its phosphorylation [93]. Subsequent studies by the same group identified a Thr-258/Ser-259 trafficking motif associated with substance P-mediated activation of neurokinin 1 receptor (NK1R)-induced NET downregulation and phosphorylation [94]. Although Ser-259 is the predicted potential PKC site, phosphoamino acid analysis revealed the presence of both phospho-Ser and phospho-Thr residues following PKC activation [94]. Mutating both Thr-258 and Ser-259 sites to non-phosphorylatable Ala prevented PKC-mediated NET phosphorylation and downregulation, while the Thr-258Ala/Ser-259Ala mutant exhibited increased basal phosphorylation, suggesting that phosphorylation of this motif may affect other phosphorylation sites [92]. Other phosphorylation sites, including Thr-19, Thr-30, Thr-58, Ser-502, Ser-579, Thr-580, and Ser-583, are not implicated in PKC-mediated NET downregulation. Therefore, PKC-mediated Thr-258 phosphorylation may play a modulatory role in NET regulation but in concert with Ser-259 phosphorylation, dictates NET endocytosis. Human and animal studies have shown that cocaine upregulates NET [98,99,100,101]. Mannangatti et al., showed that NET upregulation by cocaine is mediated by enhanced phosphorylation and diminished constitutive endocytosis of NET [101]. Cocaine-mediated NET upregulation and phosphorylation are sensitive to p38 MAPK inhibition and substitution of Thr-30 phospho-site to non-phosphorylatable alanine, suggesting a critical role of Thr-30 phosphorylation in the functional regulation of hNET in response to cocaine exposure [102].

Identification of the co-existence of protein phosphatase 2A catalytic subunit (PP2Ac) and monoamine transporters as a physical complex suggests that PP2Ac plays a role in dephosphorylating phosphorylated transporters [103,104]. Therefore, modulation of transporter/phosphatase associations influences the phosphorylation status of the transporter through dephosphorylation, affecting transporter trafficking and function. In addition to NET phosphorylation, syntaxin 1A, a presynaptic protein, is known to bind the N-terminal domain of NET. When the N-terminal domain of hNET is truncated, it disrupts the association between NET and syntaxin 1A, which limits the ability of PKC to downregulate NET function [105]. Interestingly, the PKC motif in NET Thr258/Ser259 is involved in the regulation of lipid raft mediated NET interaction and translocation of NET-NK1R-PKC complexes underscoring the importance of NET phosphorylation in membrane microdomain-specific signaling pathways specific to membrane microdomains [104]. Furthermore, NET also exists in association with active p38 MAPK (phospho-p38 MAPK), and this association is enhanced by cocaine or AMPH treatment [101,102]. Cocaine is known to activate p38 MAPK and thus it is possible that cocaine-activated p38 MAPK may be involved in NET-Thr30 phosphorylation either directly or indirectly. Collectively, these studies suggest close interaction of NET with kinases and phosphatases involved in regulating NET phosphorylation. Figure 2 summarizes kinase-, receptor- and psychostimulant-mediated regulation of NET activity by phosphorylation at specific phospho-sites.

### 3.2. Role of NET Phosphorylation in Psychostimulant Use Disorders

Animals with a history of psychostimulant exposure show dysfunction in amine transporter activity, phosphorylation, and interactions with other proteins [101,106,107,108,109]. NET is regulated by cellular signals [18,94,97,110] and psychostimulants [95,99,101,111,112]. NE controls cognition, executive and non-executive functions via the cortex [113], and relapse to drug-use involves memory consolidation and impaired cognitive functions [114,115,116]. Functional deficits in medial prefrontal cortex (mPFC)-mediated cognitive control are the main driving forces for compulsive drug-seeking/taking [117,118]. The nucleus accumbens (NAc) is a pivotal brain structure in reward pathways [117]. Thus, NE’s role in mPFC and NAc functions underscores the importance of NET regulation in psychostimulant use disorders. AMPH is a substrate for both the NET and DAT with near equal affinity and regulates their function and expression [119]. While DA and DAT contribute to drug-reward [120], NET also contributes to DA dynamics modulated by AMPH and cocaine in the mPFC [121], and also in the NAc when DAT is absent or deficient by clearing DA from extracellular space as effectively as DAT [122]. Thus, increasing evidence supports the importance of NET in contributing to mPFC and NAc functions in SUD [123,124].

Psychostimulants like cocaine and AMPH bind to NET, and inhibiting NET is necessary for these psychostimulants to produce their behavioral effects [19,20,125]. Altered NE neurotransmission and presynaptic NET function and expression have long been associated with substance use disorders, and NET is blocked by cocaine. In fact, NET levels are higher in the brains of human cocaine addicts as well as in cocaine self-administering animals [98,99,100]. Jayanthi and colleagues demonstrated that the phosphorylation of NET and its interaction with p38 MAPK increase when p38 MAPK is activated after cocaine exposure [101]. Cocaine treatment of hNET-expressing human placental trophoblast cells upregulated function, surface expression, and phosphorylation of hNET, which is blocked by p38-MAPK inhibitor [101]. Additionally, cocaine treatment inhibited constitutive endocytosis of hNET. Mutational analysis of Ser and Thr residues revealed that substitution of Thr-30, located at the amino terminus of hNET with non-phosphorylatable Ala (Thr30Ala-hNET), abolished cocaine-induced upregulation of NET function, surface expression, and phosphorylation. This suggests that the phosphorylation of NET at Thr-30 plays a crucial role in the functional regulation of hNET in response to cocaine. In vivo administration of a p38 MAPK inhibitor (SB203580) or membrane permeable TAT-NET-Thr30 peptide encompassing Thr-30 motif but not TAT-NET-Thr30Ala (mutant peptide) completely blocked cocaine-mediated NET upregulation and phosphorylation [102]. In the cocaine conditioned place preference (CPP) paradigm, mice that received TAT-NET-Thr30 but not TAT-NET-Thr30Ala exhibited significant reduction in cocaine CPP (reward) on the post-conditioning test day. It is known that cocaine binds to monoamine transporters and blocks reuptake of monoamines, resulting in increased synaptic amines. NET upregulation is a neuroadaptation occurring as a consequence of having been exposed to cocaine, and this will have significant effects on NE signaling downstream of adrenergic receptors and perhaps on DA signaling since NET is known to clear DA in PFC and when DAT is deficient [121,122,126], which in turn drives compulsive drug-seeking and/or relapse to drug-seeking/taking behaviors. It is possible that during cocaine-free periods (abstinence/withdrawal), the upregulation of NET that results from prior cocaine use may enhance the clearance of synaptic NE. This would allow for phasic releases of NE and DA in response to drug-related stimuli by reducing autoreceptor control. In other words, NET upregulation takes off the brakes on autoreceptor-mediated inhibition of NE/DA release, triggering phasic NE/DA release in response to stimuli (drug or cue) making the subject more susceptible to seeking drug or relapsing into drug-seeking behavior.

Jayanthi et al. have identified Thr258 and Ser259 residues in NET as a key motif for phosphorylation of NET by PKC, which in turn, is a critical mode for NET regulation [93,94]. Studies from the same group showed that PKC activation downstream of NK1R activation phosphorylates Thr258/Ser259 motif driving NET downregulation [93,94]. It is well-established that the monoamine transporters including NET are downregulated by the psychostimulant, AMPH [95,111]. While there is no direct evidence to show if AMPH regulates NET phosphorylation status, studies by Dipace et al., demonstrated that AMPH mediates CaMKII-dependent NET downregulation linked to NET-syntaxin 1A complex formation [111]. Interestingly, Annamalai et al., found that the AMPH-mediated NET downregulation is blocked when the PKC motif Thr258/Ser259 in NET is mutated to Ala residues suggesting that Thr258/Ser259 is required for AMPH-mediated NET downregulation [94,95]. In addition, in vivo studies by Mannangatti et al., demonstrated that the Thr258/Ser259-dependent compromised NET activity contributes to AMPH-mediated behaviors [127]. Since NE controls memory and cognition [113] and relapse to drug-use involves memory consolidation and impaired cognitive functions [114,115,116], interest in developing pharmacotherapies targeting NE transmission in the treatment of neuropsychiatric disorders is strong [113,128,129,130]. Moreover, NE-mediated mPFC and NAc functions underscores the importance of NET regulation in these brain regions. Therefore, the study by Mannangatti et al., showing that NAc specific interruption of Thr258/Ser259-dependent NET phosphorylation and regulation attenuates both expression and reinstatement of AMPH seeking behaviors in rats [127] emphasizes the existence of a causal relationship between Thr258/Ser259-dependent NET downregulation and psychostimulant actions including animal behavior. However, understanding the role of the NET-Thr258/Ser259 motif in AMPH-mediated NET regulation [95] and in AMPH-induced behaviors that index rewarding properties of drugs [127] is of importance. Therefore, recently reported transgenic knock-in (KI) mouse carrying a NET-Thr258Ala/Ser259Ala mutation [131] is a promising step forward in understanding the neurobiological role of NET phosphorylation in general, and NET-Thr258/Ser259 phosphorylation in particular, from a behavioral perspective in a living animal. The NET-Thr258Ala/Ser259Ala homozygous knock in mice showed genotype-, sex-, and brain region-specific changes in NET functional expression as well as genotype- and sex-specific differences in AMPH behavioral response compared to its wild-type littermates [131]. This study demonstrated that the NET-Thr258Ala/Ser259Ala mice exhibit brain region-specific differential expression of NET, and NET function in PFC and NAc, the regions important for drug reward. Both male and female NET-Thr258Ala/Ser259Ala mice have reduced NET function and expression in the PFC, and intact NET function and expression in the NAc when compared to their WT littermates. These findings from NET-Thr258Ala/Ser259Ala mice recapitulate their findings from in vitro studies where mutated Thr258Ala/Ser259Ala-hNET showed reduced NE transport, and reduced total and surface NET expression compared to WT-hNET [94]. 

It is well-known that AMPH reduces both NET function and its surface expression [95,111]. It is also known that manipulating the Thr258/Ser259 motif in male rats eliminates AMPH-induced NET downregulation and related behaviors. [127]. In line with these studies, the study by Ragu Varman et al., showed that AMPH-induced NET downregulation is significantly reduced in male NET-Thr258Ala/Ser259Ala mice and these mice exhibit resistance to behaviors evoked by AMPH such as psychomotor activation and reward [131]. It is important to note that AMPH-mediated DAT downregulation in terms of DAT function and DAT surface expression was intact in these mice [131]. While these observations underscore the importance of Thr258/Ser259-dependent in vivo NET regulation in psychostimulant-evoked behaviors, it is intriguing to note the reduced sensitivity to the behavioral effects of AMPH was not apparent in females at the doses tested (1.0 mg/kg for locomotor activity, 0.5 mg/kg for CPP). Further investigations in terms of response to different doses of AMPH along with investigation of effects of sex-specific hormones is warranted because AMPH’s dose–response curves to induce these behaviors are often biphasic [132,133,134] and there are sex-specific differences in the responses to psychostimulants [135,136,137,138,139,140,141], Nonetheless, collectively, findings from these studies indicate a strong causal mechanistic link between Thr258/Ser259-phosphorylation-mediated NET downregulation and AMPH behaviors.

## 4. Discussion and Conclusions

Depression and substance use disorders affect millions of individuals and place a heavy burden on the healthcare system, economy, productivity, and quality of life. While drugs that block the SERT or NET elicit antidepressant effects and are clinically proven treatment options for depression, they show only a 50% response rate and are associated with unwanted side effects. Similarly, both SERT and NET contribute to psychostimulant SUD, but there are no FDA-approved drugs available to treat SUD. Therefore, gaining new fundamental neurobiological insights into the causal mechanisms underlying the affective aspects of major depression and psychostimulant use disorder is essential for developing new treatment strategies. As reported in this review, our understanding of the role of endogenous SERT or NET phosphorylation in regulating 5-HT or NE clearance and the correlation of changes in synaptic 5-HT or NE to the changes in animal behavior are just beginning to emerge. Studies from recently developed mouse models carrying phospho-lacking monoamine transporters demonstrate the validity of targeting phosphorylation-specific transporters to gain knowledge on the role of in vivo transporter phosphorylation in animal behavior [84,131,142,143]. Table 1 summarizes the regulatory mechanisms mediated by the phosphorylation of SERT and NET linked to specific phospho-sites in these transporters, suggesting that phosphorylation acts as a molecular switch controlling transporter conformation, trafficking, and drug responses by regulating synaptic amines and subsequent aminergic neurotransmission and behaviors. Since both 5-HT and NE play essential roles in addiction and mood and modulate DA signals, striking a balance between these signals appears to be an important factor in normal physiology. Thus, changes in the phosphorylation of SERT and NET, and hence changes in their functional capacity, may disrupt this optimum balance between these monoamines, leading to neuropsychiatric conditions and behaviors mediated by psychostimulants. Detailed phosphorylation map of monoamine transporters in response to various stimuli, including drugs of abuse, along with the identification of protein phosphatases to reverse the phosphorylation events, will significantly advance our understanding of monoamine transporter regulation and potentially lead to the identification of novel therapeutics to treat addiction and other neuropsychiatric disorders.

**Table 1 ijms-26-07713-t001:** Regulation of SERT and NET Phosphorylation and the effect of Kinases, Phosphatases, Receptor ligands, and transporter substrates and inhibitors.

Regulators	Effects on Transporter Function and Trafficking	Effects on Transporter Phosphorylation	References
Protein kinases:	SERT: Reduces 5-HT uptake, SERT Vmax, and surface SERT while increasing SERT endocytosis. It exhibits biphasic effects in platelets: In the initial phase (5 min), it inhibits 5-HT uptake, reduces Vmax, and lowers 5-HT affinity without affecting surface SERT. The later phase (30 min) continues to inhibit 5-HT uptake and reduce Vmax without altering 5-HT affinity, while enhancing SERT internalization.	Increases SERT phosphorylation During the initial phase, SERT is phosphorylated on Ser residues, followed by phosphorylation on both serine and threonine residues at later phase. Phosphorylates Ser149, Ser277 and Thr603 sites in vitro hSERT-peptide phosphorylation assay.	[41,47,64]
PKC-activation	NET: Decreases NE uptake, NET Vmax, and surface NET while increasing NET endocytosis.	Increases NET phosphorylation. Phosphorylates Thr258 Ser259 sites.	[93]
p38 MAPK-inhibition	SERT: Decreases 5-HT uptake, SERT Vmax, and Km. with or without changes in surface SERT proteins.	Decreases SERT phosphorylation. Phosphorylation of Thr616 by in vitro hSERT-peptide phosphorylation assay.	[46,64]
	NET: Increases NE uptake and NET surface expression. p38 MAPK associates with NET in the presence of cocaine.	Involved in basal phosphorylation of NET. Blocks cocaine-induced NET phosphorylation.	[101,102]
PKG-activation	SERT: Increases 5-HT uptake, SERT Vmax with no effect on 5-HT affinity Km. Trafficking dependent and/or independent.	Phosphorylation of theThr276 site in hSERT is required for PKG to stimulate SERT.	[45]
CaMKII- Inhibition	SERT: Decreases 5-HT uptake.	Decreases SERT phosphorylation Phosphorylated Ser13 in in vitro hSERT-peptide phosphorylation.	[41,59,64]
Src-tyrosine kinase-activation/inhibition	SERT: Increases 5-HT uptake, SERT Vmax, surface expression, and stability, while inhibition produces the opposite effect.	Phosphorylation of Tyr47 and Tyr142 in hSERT is required for Src-Induced increases in 5-HT uptake and SERT stability.	[66]
GSK3ß-activation/inhibition	SERT: Reduces SERT function, Vmax, surface density, and the opposite with inhibition.	Phosphorylation of Ser48 in hSERT is required for GSK3ß-mediated regulation of SERT function and trafficking.	[53]
PKA activation	SERT: No effect on SERT activity.	Increases SERT phosphorylation.	[41]
Akt/PKB inhibition	SERT: Reduces SERT function, Vmax, surface density, SERT exocytosis.	Decreases SERT phosphorylation.	[44]
Phosphatases: PP2Ac-inhibition	SERT: Decreases 5-HT uptake.	Increases SERT phosphorylation. Associates with SERT.	[41]
Receptor ligands: H_3_R-agonists	SERT: Reduces 5-HT uptake, SERT Vmax, with no effect on Km, and decrease surface expression.	Decreases SERT phosphorylation.	[52]
NGF	SERT: Increases 5-HT uptake.	Increases Ser phosphorylation of SERT.	[60]
KOR-agonists	SERT Reduces 5-HT uptake, SERT Vmax with no effect on Km, and surface expression.by reducing exocytosis and increasing endocytosis	Increases SERT phosphorylation.	[67]
NK1R-agonists	NET: Decreases NE uptake, NET Vmax, and surface NET while increasing NET endocytosis.	Increases NET phosphorylation via PKC activation. Thr258 and Ser259 sites are required for NK1R-mediated NET phosphorylation and raft-mediated subcellular translocation.	[94,104]
Transporter substrates: 5-HT	SERT: Upregulates surface SERT and attenuates PKC-dependent surface down regulation.	Attenuates PKC-dependent SERT phosphorylation.	[43]
AMPH	SERT: Not known.	Increases SERT phosphorylation through p38 MAPK pathway.	[46]
	NET: Downregulates NET and Thr258/Ser259 PKC site is required.	Not known.	[95,127]
Fenfluramine	SERT: Attenuates PKC-dependent surface down regulation.	Attenuates PKC-dependent SERT phosphorylation.	[43]
Transporter inhibitors: Paroxetine, Citalopram, Imipramine, and Cocaine	SERT: Attenuate PKC-dependent surface down regulation.	Attenuates PKC-dependent SERT phosphorylation.	[43]
Cocaine	NET: Increases NE uptake and upregulates surface NET and is blocked by p38 MAPK inhibition.	Increases NET phosphorylation which is blocked by p38 MAPK inhibition. Thr30 is required for cocaine-mediated NET phosphorylation.	[101,102]

## 5. Future Directions

Phosphorylation-dependent regulation of SERT or NET is a dynamic process, involving regulated transporter trafficking/recycling affecting these neurotransmitter transporter functions and consequently the synaptic dynamics of 5-HT and NE involving many behaviors associated with neuropsychiatric disorders [36,41,46,47,56,66,93,101,102,127,131]. Studies are just emerging reporting whether phosphorylation of SERT and NET directly impacts transporter function/trafficking in vivo and whether it mediates 5-HT and NE linked behaviors. While studies report altered SERT phosphorylation in OCD and ASD associated with SERT gene coding variants (which are not actual kinase sites) and link SERT phosphorylation to human disease states, such studies are lacking that link NET phosphorylation with disease-associated NET coding variants. However, correlational analyses of psychostimulant-mediated behavioral difference scores with psychostimulant effects on NET regulatory phenomena and on NET-mediated NE/DA dynamics in NET-T258A/S259A mice will allow integrating the changes in regulatory mechanisms occurring due to transporter phosphorylation into the actions of these psychostimulants including animal behavior. While studies using animal models of SERT or NET phosphorylation are just beginning to come to light, there exists a large gap in transferring the knowledge from animal studies to humans or post-mortem tissues from patients diagnosed with neuropsychiatric conditions. Such studies will be a leap forward in the field.

Several phosphorylation sites exist in SERT and NET, and the identity of the kinases and their roles in transporter regulation linked to human disease states need to be explored further with an emphasis on investigating potential therapeutic benefits of targeting transporter phosphorylation. More importantly, delineating auto- and heteroreceptors located in serotonergic and noradrenergic neurons, along with their downstream signaling cascades that regulate kinase/phosphatase-mediated SERT and NET transporter phosphorylation will enhance our understanding of the circuit interactions with neuronal factors regulating neurotransmission and behaviors. Such studies potentially can reveal whether dysregulated SERT and NET phosphorylation and the consequent altered 5-HT and NE neurotransmission predispose to psychiatric disorders and SUD. Detailed phosphorylation map of monoamine transporters in response to various stimuli, including drugs of abuse, along with the identification of protein phosphatases to reverse the phosphorylation events, will significantly advance our understanding of monoamine transporter regulation and potentially lead to the identification of novel therapeutics to treat addiction and other neuropsychiatric disorders. Future studies developing and utilizing in vivo phosphorylation assays and kinase/phosphatase knockout animal models are warranted to map transporter phosphorylation completely. Once fully mapped, one can envision the development of targeted small molecule therapies that affect site-specific phosphorylation of specific transporters. These may include CNS-penetrant receptor, kinase and phosphatase modulators as well as minigenes capable of expressing peptides that intervene site-specific phosphorylation of monoamine transporters. The phosphorylation-specific targeted approaches have the potential to improve treatments for mood and other neuropsychiatric disorders.

## Figures and Tables

**Figure 1 ijms-26-07713-f001:**
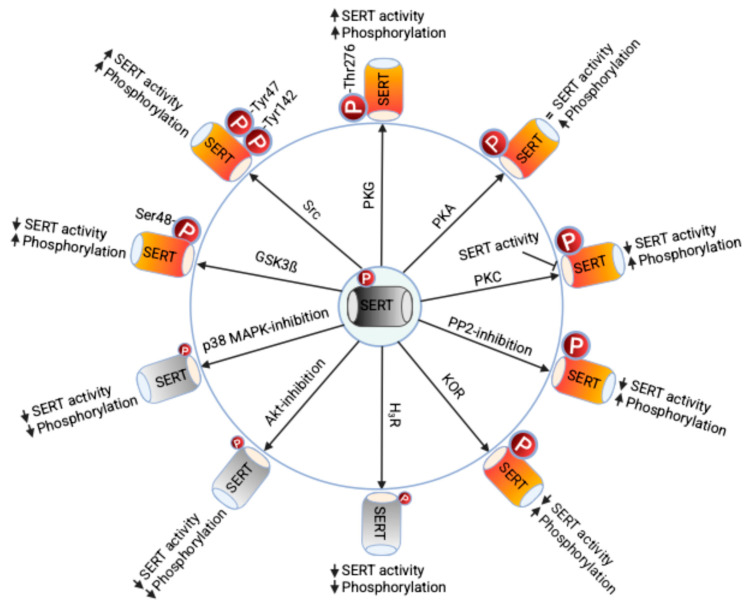
Regulation of SERT phosphorylation and function. SERT is constitutively phosphorylated and exists as a phosphoprotein. As shown in the cartoon, the activation and inhibition of several protein kinases, receptors, and phosphatase activity can alter SERT’s basal phosphorylation and regulate its intrinsic transport properties. Additionally, phosphorylation affects SERT levels at the plasma membrane by influencing the recycling processes involved in endocytosis and exocytosis. Specifically, Tyrosine phosphorylation of SERT at sites 47 and 48 governs SERT protein degradation and its expression on the cell surface. The activation of PKG enhances SERT activity by phosphorylating Thr276 in SERT without affecting surface levels SERT. Phosphorylation of Ser48 in SERT by active GSK3ß inhibits SERT function by modulating its endocytosis and exocytosis. The causal relationship between SERT phosphorylation and regulation mediated by Akt, PKC, p38 MAPK, H3R, and KOR remains unclear (refer to Table 1). Therefore, further investigation into a comprehensive phosphorylation map of SERT in response to various stimuli, including antidepressants, drugs of abuse, and disease-linked SERT coding variants, along with identifying protein phosphatases responsible for reversing phosphorylation events, will significantly enhance our understanding of the role of SERT phosphorylation in serotonergic neurotransmission in both normal and disease states. Constitutively phosphorylated SERT is represented in dark grey while reduced SERT phosphorylation in lighter grey and enhanced SERT phosphorylation in red-orange.

**Figure 2 ijms-26-07713-f002:**
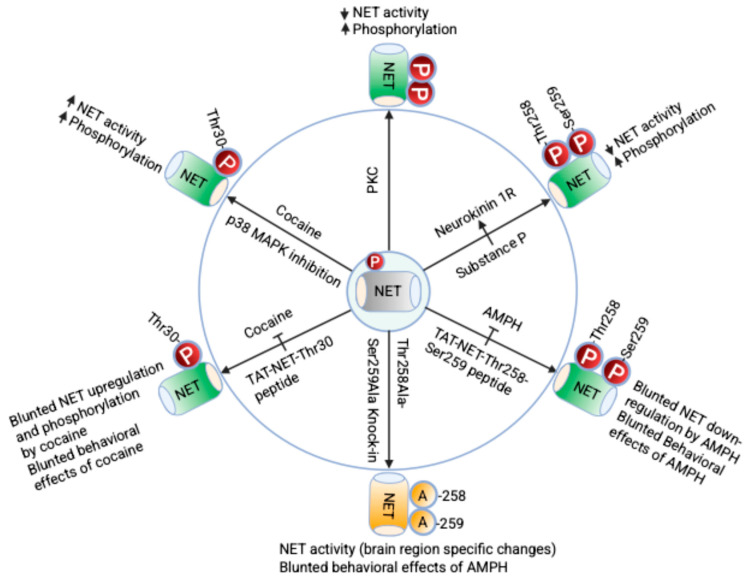
Regulation of NET phosphorylation and function. NET is constitutively phosphorylated and exists as a phosphoprotein. As shown in the cartoon, the activation and inhibition of protein kinases, and receptors as well as psychostimulants like cocaine can alter basal phosphorylation of NET and regulate its intrinsic transport properties. Additionally, phosphorylation affects plasma membrane expression of NET by influencing its endocytosis and exocytosis. Specifically, PKC activation enhances NET phosphorylation and decreases its function by decreasing its expression on the cell surface. When NK1R is activated by its agonist, substance P (SP), protein kinase C (PKC) is activated, which in turn phosphorylates NET at the Thr258/Ser259 sites. Blocking NET-Thr258/Ser259 phosphorylation in vivo by targeting the NET-Thr258/Ser259 motif using the membrane-permeable TAT-NET-Thr258/Ser259 peptide blunts AMPH-mediated NET downregulation and AMPH-induced hyperlocomotion and reward like CPP behaviors. In NET-Thr258Ala/Ser259Ala knock-in mice, which express a non-phosphorylatable alanine substitution at the Thr258 and Ser259 sites in NET, a phenotype emerges. These mice show resistance to the downregulation of NET by AMPH. Additionally, they display decreased locomotor activity and a reduced reward response when exposed to AMPH. Additionally, cocaine exposure and its binding to NET activates p38 MAPK pathway through mechanisms that are not yet fully understood. Activated p38 MAPK phosphorylates NET at Thr30 site, which reduces NET endocytosis, thereby increasing surface NET and NE uptake. Blocking NET-Thr30 phosphorylation through TAT-NET-Thr30 peptide not only suppresses cocaine-induced elevated NET function and surface expression but also prevents the behavioral effects of cocaine, such as locomotor sensitization, reward, and reinstatement. Thus, developing reagents that target specific sites or motifs phosphorylated in NET will further our understanding of the role of NET phosphorylation in noradrenergic neurotransmission contributing to both mood disorders and SUD. Constitutively phosphorylated NET is represented in grey while enhanced NET phosphorylation is represented in green and NET KI mouse model carrying Ala at amino acid positions 258 and 259 is represented in yellow.

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
