# Peer review of "Role of Phosphorylation of Serotonin and Norepinephrine Transporters in Animal Behavior: Relevance to Neuropsychiatric Disorders"

_ijms, 2025, doi:10.3390/ijms26167713_

Round 1

Reviewer 1 Report

Comments and Suggestions for Authors

Jayanthi and Ramamoorthy prepared a comprehensive review article that dissects the mechanistic details and functional outcomes of SERT and NET phosphorylation in vitro and in vivo. The article effectively synthesizes research findings over the past three decades on this important topic. However, a major concern is the amount of text overlap with prior work flagged in the iThenticate report. In several instances, entire sentences or even full paragraphs appear to be duplicated from previously published work. Therefore, the manuscript will require extensive paraphrasing.

Other suggestions:

- consider including at least one recent example of CNS-penetrant kinase modulators as potential treatment options (https://doi.org/10.1073/pnas.1809137115)

- consider adding a figure analogous to Figure 1 to summarize NET regulation

Minor edits:

Lines 124-127: “Unlike PKC-dependent downregulation of SERT, which decreases surface SERT levels and increases SERT phosphorylation, p38 mitogen-activated protein kinases (p38 MAPK) enhance SERT activity and surface SERT, as well as promote SERT phosphorylation suggests that PKC phosphorylates SERT at distinct sites to regulate its function and trafficking” - this sentence is missing words and/or punctuation

Line 345: "Inturn" should be "in turn"

Line 350: "breaks" should be "brakes"

Reviewer 2 Report

Comments and Suggestions for Authors

The manuscript provides a comprehensive review of the role of serotonin (SERT) and norepinephrine (NET) transporter phosphorylation in animal behavior and its relevance to neuropsychiatric disorders. The authors effectively summarize current knowledge, highlighting the dynamic regulation of these transporters by phosphorylation and its implications for mood disorders and psychostimulant use. The review is well-structured, with clear sections and logical flow. However, some areas could benefit from further clarification, expansion, or critical analysis.

Major Comments

    • The manuscript thoroughly describes phosphorylation sites and their regulatory roles but could better delineate the specific kinases and phosphatases involved in each context. For example, Table 1 is helpful but could be expanded to include more details about the downstream effects of phosphorylation (e.g., conformational changes, protein-protein interactions).
    • Suggest adding a schematic figure summarizing the phosphorylation sites, kinases, and functional outcomes for SERT and NET to enhance clarity.
    • While the link between transporter phosphorylation and animal behavior is discussed, the translation to human neuropsychiatric disorders remains somewhat speculative. The authors should emphasize gaps in knowledge, such as whether phosphorylation changes observed in animal models are replicated in human studies or post-mortem tissues.
    • The discussion of SERT phosphorylation in autism spectrum disorder (ASD) and obsessive-compulsive disorder (OCD) is intriguing but would benefit from more critical analysis of conflicting or inconclusive findings in human genetic studies.
    • The section on cocaine and amphetamine regulation of NET phosphorylation is well-detailed, but the mechanistic link to addiction behaviors could be strengthened. For example, how does NET upregulation after cocaine exposure specifically contribute to relapse or craving in animal models?
    • The role of NET in dopamine clearance in the prefrontal cortex is mentioned, but its broader implications for reward circuitry could be expanded.
    • The "Future Directions" section is somewhat generic. The authors could propose specific experiments (e.g., in vivo phosphorylation assays, kinase knockout models) to address unresolved questions.
    • A discussion of potential therapeutic strategies targeting phosphorylation (e.g., kinase inhibitors) would be valuable, especially given the limited efficacy of current antidepressants.

Minor Comments

    • Some abbreviations (e.g., PKG, CaMKII) are defined late in the text or not at all. Ensure all abbreviations are defined at first use.
    • The term "phospho-dynamics" is used but not clearly defined. Consider clarifying or replacing with "phosphorylation dynamics."
    • Figures 1 and 2 are referenced but not included in the provided text. Ensure these are clearly labeled and described in the manuscript.
    • Table 1 is comprehensive but could be formatted for better readability (e.g., grouping kinases by family).
    • Some statements lack citations (e.g., "NE controls arousal, mood, attention..."). Ensure all claims are supported by references.
    • Check for consistency in citation style (e.g., "Jayanthi et al., 2005" vs. "Jayanthi and Ramamoorthy, 2005").
    • Minor grammatical errors exist (e.g., "inturn" should be "in turn"). A thorough proofread is recommended.
    • Some sentences are overly long and could be simplified for readability (e.g., "The contribution of SERT and NET to the regulation of neurotransmitters 5-HT and NE, to disease processes and that they are a target of several drugs of abuse highlight the importance...").

Round 2

Reviewer 1 Report

Comments and Suggestions for Authors

The authors have fully addressed the concerns raised during round 1 review and incorporated all suggestions. 

Author Response

Comment: The authors have fully addressed the concerns raised during round 1 review and incorporated all suggestions.

Response: We thank the reviewer for their time and suggestions on round one review. We thank the reviewer again.

Reviewer 2 Report

Comments and Suggestions for Authors

While the article is well-developed, there are certain areas that would benefit from clarification, refinement, and better integration of key concepts. Below are specific comments and suggestions categorized as minor revisions.

Minor Comments

  • Some abbreviations appear before being defined (e.g., NAc, NK1R). Ensure all abbreviations are defined at first mention and consistently used thereafter.
  • A few grammatical inconsistencies and overly long sentences impede readability. For instance:

“However, the neurobiological consequences and behavioral outcome of in vivo regulation of SERT and NET by phosphorylation are not fully understood.”

  • Suggestion: Break complex sentences into simpler structures where appropriate to enhance clarity.
  • This is a valuable resource but could benefit from column alignment adjustments and uniform phrasing.
  • Add missing reference numbers and ensure consistent citation formatting.
  • The conclusion rightly emphasizes the translational potential but could better articulate the therapeutic implications of manipulating transporter phosphorylation, including existing small molecules or drug candidates.

Author Response

We thank the reviewer for their positive comments on the article. Reviewer also indicated need for improvement in their Minor Comments, which we have addressed in this 2nd revision. Responses in blue font.

  • Some abbreviations appear before being defined. We have updated abbreviations at the first appearance as suggested.
  • A few grammatical inconsistencies and overly long sentences. We have broken down long sentences to reduce grammatical inconsistencies and for better readability.
  • Add missing reference numbers and ensure consistent citation formatting. We have updated missing references and used correct formatting.
  • The conclusion could better articulate the therapeutic implications of manipulating transporter phosphorylation. We have expanded on our view on therapeutic developments in the future directions as suggested.

We thank the reviewer again for their constructive criticisms and hope our responses satisfactorily addressed them.